# The Value of Deep Learning in Gallbladder Lesion Characterization

**DOI:** 10.3390/diagnostics13040704

**Published:** 2023-02-13

**Authors:** Yunchao Yin, Derya Yakar, Jules J. G. Slangen, Frederik J. H. Hoogwater, Thomas C. Kwee, Robbert J. de Haas

**Affiliations:** 1Department of Radiology, Medical Imaging Center Groningen, University Medical Center Groningen, University of Groningen, 9700 RB Groningen, The Netherlands; 2Department of Surgery, Section Hepato-Pancreato-Biliary Surgery and Liver Transplantation, University Medical Center Groningen, University of Groningen, 9700 RB Groningen, The Netherlands

**Keywords:** gallbladder, deep learning, artificial intelligence, cancer

## Abstract

Background: The similarity of gallbladder cancer and benign gallbladder lesions brings challenges to diagnosing gallbladder cancer (GBC). This study investigated whether a convolutional neural network (CNN) could adequately differentiate GBC from benign gallbladder diseases, and whether information from adjacent liver parenchyma could improve its performance. Methods: Consecutive patients referred to our hospital with suspicious gallbladder lesions with histopathological diagnosis confirmation and available contrast-enhanced portal venous phase CT scans were retrospectively selected. A CT-based CNN was trained once on gallbladder only and once on gallbladder including a 2 cm adjacent liver parenchyma. The best-performing classifier was combined with the diagnostic results based on radiological visual analysis. Results: A total of 127 patients were included in the study: 83 patients with benign gallbladder lesions and 44 with gallbladder cancer. The CNN trained on the gallbladder including adjacent liver parenchyma achieved the best performance with an AUC of 0.81 (95% CI 0.71–0.92), being >10% better than the CNN trained on only the gallbladder (*p* = 0.09). Combining the CNN with radiological visual interpretation did not improve the differentiation between GBC and benign gallbladder diseases. Conclusions: The CT-based CNN shows promising ability to differentiate gallbladder cancer from benign gallbladder lesions. In addition, the liver parenchyma adjacent to the gallbladder seems to provide additional information, thereby improving the CNN’s performance for gallbladder lesion characterization. However, these findings should be confirmed in larger multicenter studies.

## 1. Introduction

The diagnosis of gallbladder cancer (GBC) remains a challenge in clinical practice because of its similarity to benign gallbladder disease. Therefore, GBC is often diagnosed at a relatively late stage, resulting in a poor prognosis with a five-year overall survival rate being up to only 13% [1,2,3,4]. However, when GBC is detected at an early stage, radical resection can be an option (especially in patients with T1b/T2 tumors), increasing the survival rate to 53% [4]. Besides, adequate characterization of gallbladder lesions is also important to correctly select patients that should be treated at specialized hepatobiliary hospitals.

A recent study evaluating radiologists’ performance in differentiating gallbladder lesions based on CT images achieved a high sensitivity of 90%. However, the specificity was merely 60% [5]. Another recent study adopted the quantitative approach of radiomic analysis to evaluate gallbladder lesions [6]. Various machine-learning models were built based on extracted radiomic features to differentiate GBC and benign gallbladder lesions. The specificity of the radiomic analysis achieved 80%, but the sensitivity was merely 64% [6]. In addition, when including both the gallbladder and adjacent liver parenchyma in the radiomic analysis, the diagnostic performance did not significantly improve. The best results were obtained when combining CT-based radiomics with visual radiological assessment [6]. 

Convolutional neural networks (CNNs) have shown their strong ability in medical image classification during recent years. Compared with machine learning models using extracted radiomic features as input, CNNs use all CT image information as input and exploit useful information from the image for a specific task during model training. The use of all available CT information can possibly improve the differentiation between GBC and benign gallbladder disease. 

The primary aim of this study was to determine whether a CNN can adequately differentiate GBC from benign gallbladder diseases on CT scans. The secondary aim was to investigate whether a CNN can exploit information from adjacent liver parenchyma to improve the performance of CT-based gallbladder lesion characterization. 

## 2. Materials and Methods

### 2.1. Study Population

All patients referred to our hospital (which is a tertiary referral center) between January 2007 and October 2020 with suspicion of GBC or because of an incidentally found GBC after cholecystectomy were included in the study. Exclusion criteria were the absence of a contrast-enhanced portal venous phase CT scan (for incidentally found GBC, the CT had to be performed prior to a cholecystectomy) and lacking a histopathological diagnosis confirmation. Reasons for suspicion of GBC and subsequent referral to our hospital were a polyp with a diameter > 10 mm, a focal or diffuse wall thickening without obvious signs of benign disease, a mass lesion, or a porcelain gallbladder that has been considered to increase the risk of GBC [1]. Although the CT systems were of multivendor origin, scan parameters were harmonized between our hospital and the surrounding referring hospitals (more specifically automatic tube current modulation and tube voltage selection, slice thickness of 1 mm, and a delay of 75 s after IV injection of 90–100 mL of contrast medium at a flow rate of 3.6–4.0 mL/s followed by 32 mL of saline solution). All patients were identified in a prospectively maintained surgical institutional database or by searching multidisciplinary team meeting lists and analyzed retrospectively. Approval from the institutional review board was obtained, and the need for written informed consent was waived. The current study population was also part of two previous studies which were focused on different research questions [5,6].

Collected data included: patient age, gender, date and type of surgery, date of CT, and histopathology results. In each case of GBC suspicion, an open radical cholecystectomy was performed combined with frozen section biopsy of the gallbladder. If the frozen section biopsy was positive and without signs of disseminated disease, a lymph node dissection of the hepatoduodenal ligament and a wedge resection of the gallbladder bed were performed. A similar approach was used for patients referred with an incidentally found GBC after previous cholecystectomy in the referring hospital and after excluding disseminated disease on a postoperative CT scan. Each resection and biopsy specimen underwent routine histopathological examination performed by a specialized hepatobiliary pathologist.

### 2.2. Image Processing and Deep Learning Models

The workflow of deep learning for gallbladder disease characterization is shown in Figure 1.

The 3D portal venous phase CT scans were pre-processed before being fed to the deep learning model. To improve contrast among abdominal organs, the CT scans were processed by a soft window centering at 50 HU with a width of 400 HU. To normalize the CT scans throughout the entire dataset, the images were resampled to the same spacing of 1.0, 1.0, and 2.0 by a linear interpolator. Using the software ITK-SNAP, the gallbladder on the CT scans was manually delineated by an abdominal radiologist who was blinded to the final diagnosis. Examples of the CT scans with segmented GBC and benign gallbladder disease can be found in Figure 2 and Figure 3.

The processed and segmented CT scans served as input for a CNN, a type of deep learning algorithm that is well-suited for image analysis tasks, such as image classification, and thereby, predicting disease probabilities (i.e., differentiation between GBC and benign gallbladder disease on CT scans) [7]. The CNN used in the current study consisted of six convolutional blocks and three linear layers. Each convolutional block included a transposed convolutional layer, an activation function, and a batch normalization layer. 

Because GBC is a rare disease, an imbalance existed in the dataset between GBC and benign gallbladder disease. To improve the learning efficiency of the deep learning algorithm and to avoid overfitting the imbalance of the dataset, a class weight of 2.0 was assigned to the GBC images during model training.

The CNN was trained and validated using 80% of the images randomly selected from the dataset. The remaining 20% of the CT scans were used as a test set to evaluate the performance of the trained CNN. The test set remained unseen to the model during training. The model training was terminated according to the performance on a validation set that accounted for 10% of the training set. The CNN was developed based on the deep learning framework PyTorch [8] and the performance of the model was quantified by the open-source library scikit-learn 0.23.2 with Python 3.7.9 [9].

### 2.3. Deep Learning Model Based on Gallbladder and Liver Parenchyma

In a previous study, the suspicion of invasion of adjacent liver parenchyma was observed to be positively related to GBC [5]. Therefore, in addition to using only the gallbladder on CT images when training the deep learning model, a separate analysis was performed to investigate whether the combination of the gallbladder and adjacent liver parenchyma could increase the performance of the deep learning model when differentiating between GBC and benign gallbladder disease. The segmentation of a 2 cm rim of liver parenchyma adjacent to the gallbladder was automatically generated and adjusted by an experienced abdominal radiologist if necessary. The adjacent liver parenchyma was combined with the segmented gallbladder as training data for the deep learning model. Figure 4 shows examples of input CT images with segmentation of both the gallbladder and 2 cm of adjacent liver parenchyma. The deep learning model based on the combination of the gallbladder and adjacent liver parenchyma was trained and tested by the same methodology as described for the model solely based on the gallbladder.

### 2.4. Combining Convolutional Neural Network Prediction with Radiological Visual Interpretation

In a previous study, the best results for differentiation between GBC and benign gallbladder disease were observed when combining CT-based radiomic analysis with radiological visual interpretation [6]. To determine whether the results of the CNN could also improve the radiological visual interpretation, an additional analysis was performed combining the CNN prediction with radiological visual interpretation.

The radiological visual interpretation was given in a five-point scale format by two radiologists after consensus reading [5]. Subsequently, the assigned points were converted into the probability of GBC (definitely benign = 0.0, probably benign = 0.25, equivocal = 0.5, probably GBC = 0.75, and definitely GBC = 1.0). The converted probability of radiological visual interpretation and the probability predicted by the CNN were summed up with an equal weight of 0.5 as the combined probability score. In a case of a combined probability score > 0.5, the patient was considered positive for GBC.

## 3. Results

### 3.1. Study Population

A total of 127 patients fulfilled our inclusion criteria and were therefore included in the study. The patient cohort had a median age of 66 (interquartile range: 58–73). Eighty patients were female (63%), and forty-seven were male (37%). Detailed information regarding surgical treatment and histopathological examination results can be found in Table 1.

### 3.2. Convolutional Neural Network Results

Training the CNN solely on the gallbladder on CT scans yielded an accuracy rate of 0.77 (95% CI 0.70–0.85) and an area under the receiver operating characteristic (ROC) curve (AUC) of 0.71 (95% CI 0.58–0.88) in the randomly split test set for GBC and benign gallbladder disease differentiation. By adding the adjacent liver parenchyma to the gallbladder on the CT scan, the AUC increased by >10% to 0.81 (95% CI 0.71–0.92; *p* = 0.09). The sensitivity also increased from 56% to 67% (95% CI 50–86%) with merely a 6% drop in specificity (*p* = 0.15). More detailed results are summarized in Table 2, and the ROC curves are provided in Figure 5. 

Adding the radiological visual assessment to the results of the CNN trained solely on the gallbladder, as well as to a combination of the gallbladder and adjacent liver parenchyma, did not improve the diagnostic performance (Table 2).

## 4. Discussion

In the current study, the hypotheses that a CNN can differentiate between GBC and benign gallbladder diseases and that a CNN can exploit valuable information from adjacent liver parenchyma to improve the performance of GBC diagnosis were tested. In our study population, consisting of 127 patients with 44 patients having GBC and 83 patients benign gallbladder disease, the CNN trained using CT scans including both the gallbladder and a rim of adjacent liver parenchyma yielded the best performance in differentiating between GBC and benign gallbladder disease. More specifically, an AUC of 0.81 and an accuracy rate of 77% were obtained. 

To our knowledge, this is the first study using a CNN for differentiation between GBC and benign gallbladder disease. Compared with radiomic analysis, a CNN uses all available CT image information from the segmented part as input. This theoretically could provide more information compared with radiomic analysis which uses only specific extracted radiomic features from the segmented area. This concept can be further underlined by the fact that the accuracy rate of the CNN was 3% better than that of the radiomic analysis in the same study population reported recently by our group [6]. However, the current study comprises a relatively small patient group, and therefore, should be considered a feasibility study serving as the first step towards a large multicenter study of the applicability of deep learning techniques in gallbladder lesion characterization. 

In a recent study conducted by our research group, a radiomic analysis for GBC and benign gallbladder disease differentiation was performed based on features extracted from the gallbladder and adjacent liver parenchyma. However, the diagnostic performance was not significantly improved compared with a radiomic analysis solely based on features derived from the gallbladder. A possible explanation could be that the parenchymal invasion of GBC into adjacent liver tissue might be too small to be reflected as a difference in textural features [6]. However, a CNN is considered to have a stronger ability to exploit information from CT scans. Perhaps this could explain that in the current study, the AUC of the CNN improved by 11% when including both the gallbladder and adjacent liver parenchyma CT scans as inputs into the CNN. 

We recently reported that when combining radiological visual interpretation with radiomic analysis of CT scans of patients with gallbladder lesions, the best diagnostic performance was achieved for differentiating between GBC and benign gallbladder disease [6]. However, the combined results of radiological visual interpretation and CNN prediction did not improve diagnostic performance in the current study. Perhaps this could be related to the relatively small study population in the current study, and this should be the subject of future research. In addition, when using larger datasets in future studies, gallbladder segmentation could be automated and quantified by deep learning algorithms. 

Due to the rarity of gallbladder cancer in daily clinical practice and the single-center study design, the CNN was trained and tested on a small population. The scale of the dataset could influence the generalization ability and limit the performance of the CNN. The utilization of a dataset that is both larger and more heterogeneous in nature has been demonstrated to result in improved sensitivity and specificity of CNN models. The significance of the improvement in AUC and sensitivity by adding adjacent liver parenchyma to the model should also be further validated by a more extensive dataset. As a result, the current study should be considered the first step towards a large multicenter study focusing on the ability of deep learning techniques to better characterize gallbladder diseases. Therefore, not only could patient care and long-term survival outcomes be improved, but also more efficient use of scarce highly specialized hepatobiliary health care resources might be obtained. 

## 5. Conclusions

A CT-based CNN shows promising ability to differentiate gallbladder cancer from benign gallbladder lesions. In addition, the CT-based CNN shows stronger ability to exploit information from the surrounding liver parenchyma for gallbladder lesion characterization compared with a previously reported CT-based radiomic analysis. Our results could serve as the first step towards large multicenter studies further improving artificial intelligence techniques to adequately characterize gallbladder diseases. 

## Figures and Tables

**Figure 1 diagnostics-13-00704-f001:**
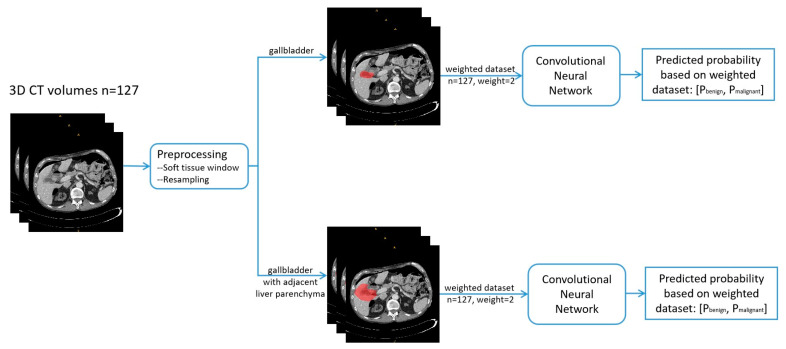
Workflow of the convolutional neural network for gallbladder cancer and benign gallbladder disease differentiation.

**Figure 2 diagnostics-13-00704-f002:**
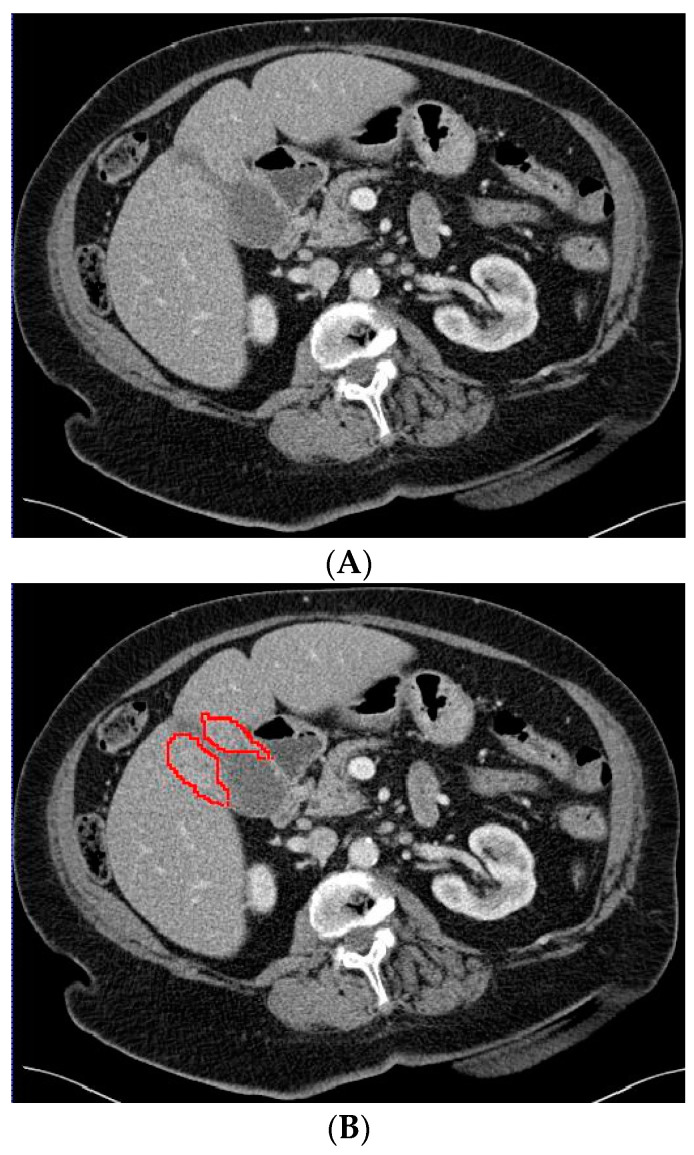
Axial CT slice (**A**) with an example of gallbladder cancer (histopathologically proven adenocarcinoma, encircled in (**B**)) and subsequent segmented gallbladder (**C**).

**Figure 3 diagnostics-13-00704-f003:**
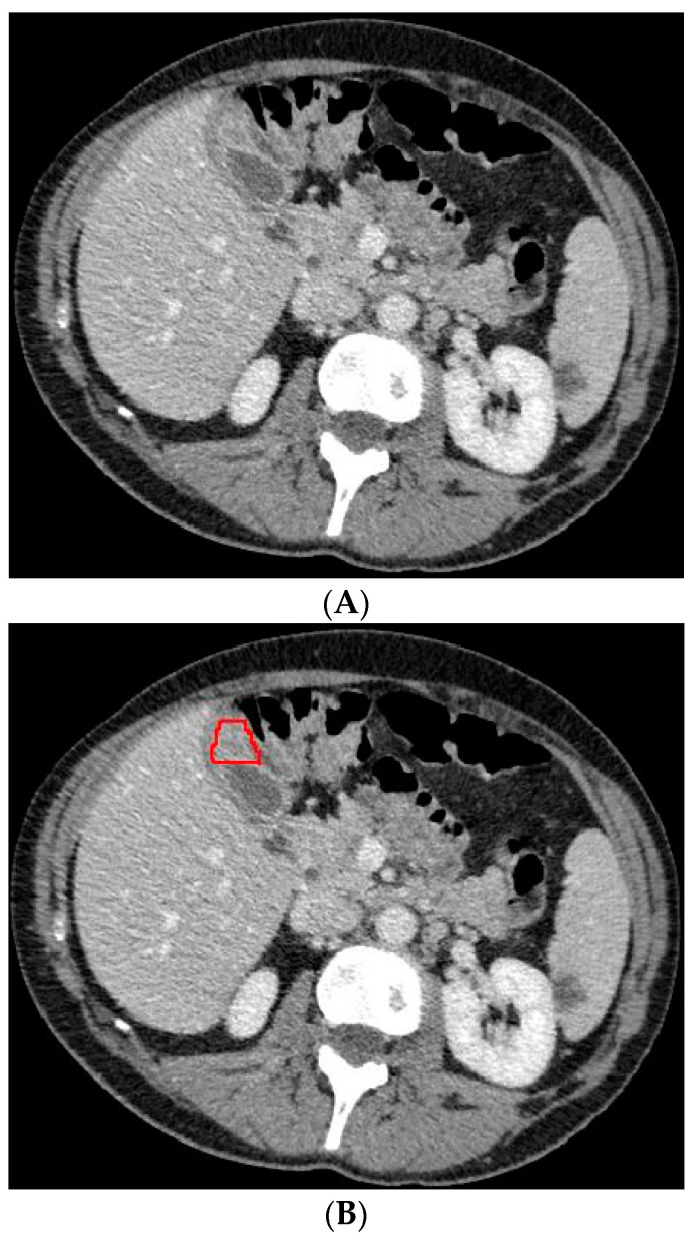
Axial CT slice (**A**) with an example of benign gallbladder disease (adenomyomatosis, encircled in (**B**)) and subsequent segmented gallbladder (**C**).

**Figure 4 diagnostics-13-00704-f004:**
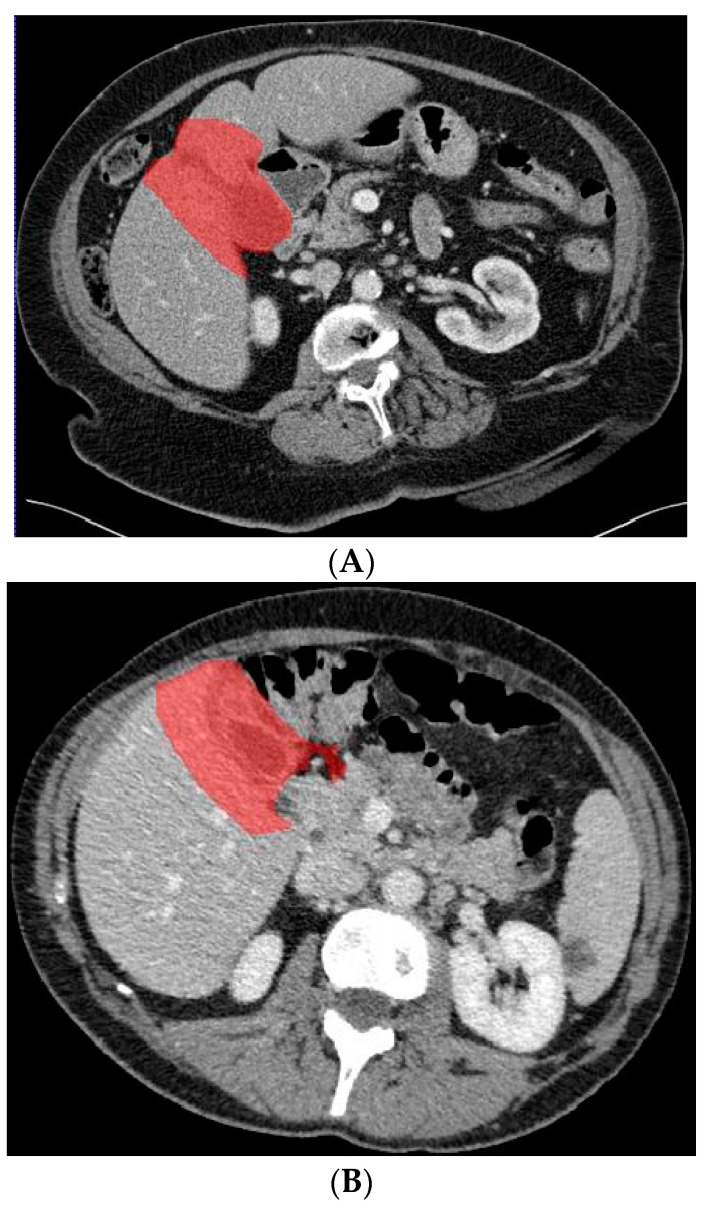
Axial CT slices with examples of segmented gallbladder including 2 cm of adjacent liver parenchyma ((**A**) gallbladder cancer case from Figure 2, and (**B**) benign gallbladder disease case from Figure 3).

**Figure 5 diagnostics-13-00704-f005:**
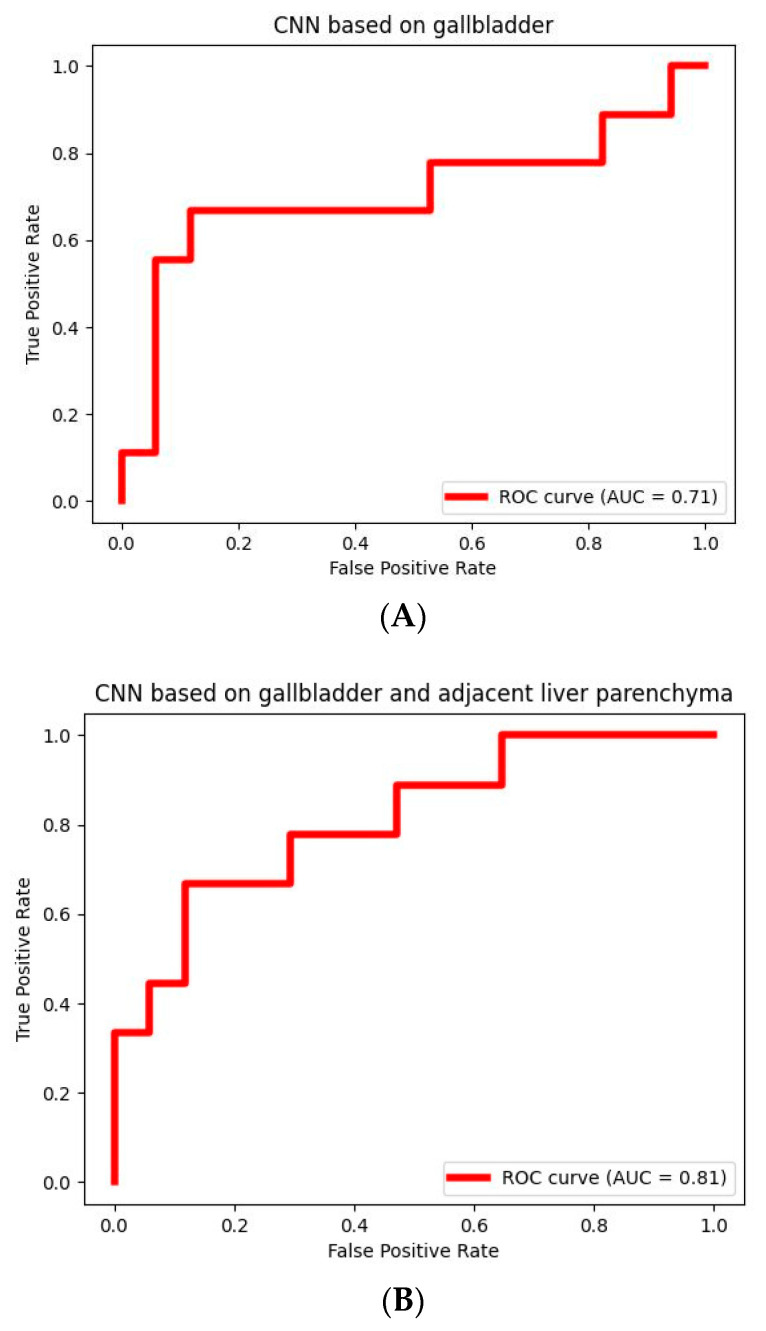
Receiver operating characteristic curves of the performance of the convolutional neural network on the test set. (**A**) CT-based convolutional neural network trained by only the gallbladder on the CT scan. (**B**) A CT-based convolutional neural network trained by the gallbladder including adjacent liver parenchyma on the CT scan.

**Table 1 diagnostics-13-00704-t001:** Histopathology results and treatment data.

Characteristic		Total Study Population N = 127
Benign gallbladder disease	Acute cholecystitis	1 (1%)
	Chronic cholecystitis	49 (39%)
	Xanthogranulomatous cholecystitis	6 (5%)
	Adenoma	4 (3%)
	Adenomyomatosis	15 (12%)
	Porcelain gallbladder	2 (2%)
	Other benign entities	6 (5%)
Gallbladder cancer	Adenocarcinoma	37 (29%)
	Adenosquamous carcinoma	3 (2%)
	High-grade dysplasia	2 (2%)
	Other types of malignancy	2 (2%)
Incidentally found gallbladder cancer		9 (7%)
Types of (surgical) treatment	Open cholecystectomy	42 (33%)
	Laparoscopic cholecystectomy	13 (10%)
	Cholecystectomy combined with resection of liver segment 4/5	5 (4%)
	Cholecystectomy combined with a wedge resection of the liver parenchyma	41 (32%)
	Cholecystectomy combined with extensive surgery *	4 (3%)
	Cholecystectomy combined with lymphadenectomy	6 (5%)
	Open-closure procedure	10 (8%)
	Biopsy without any further operation	6 (5%)

* e.g., ≥3 liver segments, and/or pancreaticoduodenectomy.

**Table 2 diagnostics-13-00704-t002:** Result of the convolutional neural network for differentiation between gallbladder cancer and benign gallbladder disease.

	AUC	Accuracy	Sensitivity	Specificity
CNN based on segmented gallbladder	0.71 [0.58, 0.88]	0.77 [0.70, 0.85]	0.56 [0.33, 0.80]	0.88 [0.83, 1.00]
CNN based on segmented gallbladder including 2 cm of adjacent liver parenchyma	0.81 [0.71, 0.92]	0.77 [0.70, 0.85]	0.67 [0.50, 0.86]	0.82 [0.75, 0.92]
CNN (only gallbladder) combined with radiological diagnosis	0.75 [0.65, 0.89]	0.73 [0.65, 0.85]	0.56 [0.40, 0.71]	0.82 [0.75, 0.93]
CNN (gallbladder including adjacent liver) combined with radiological diagnosis	0.71 [0.58, 0.86]	0.77 [0.70, 0.85]	0.67 [0.50, 0.86]	0.82 [0.75, 0.93]

Abbreviations: CNN = convolutional neural network and AUC = area under the ROC curve. Values between brackets concern 95% confidence intervals.

## Data Availability

The data presented in this study are available on request from the corresponding author.

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
