# Peer review of "The Value of Deep Learning in Gallbladder Lesion Characterization"

_diagnostics, 2023, doi:10.3390/diagnostics13040704_

Round 1

Reviewer 1 Report

The 2 tables should be supplemented with data (for instance- in how many patients the GBC was unsuspected and diagnosed only on the specimen or in how many frozen sections were performed).

It would be important to know how many patients had an incidentally found gallbladder cancer after cholecystectomy. 55 patients (43%) - had just a plain cholecystectomy (open or laparoscopic) although there was a preoperative CT available and , at least in some of these cases , a suspicion of GBC should have been raised. In an intraoperative suspicion of cancer a frozen section is the optimal attitude, avoiding the risk of overtreating benign disease with radical cholecystectomies situation but, more important, giving the patient the benefit of a single operation (radical cholecystectomy). As precise as it may be, the preoperative CT should orient only the choice of the hospital and operative team if there is a strong suspicion of GBC.

Author Response

We would like to thank the reviewer for his/her time invested in reviewing our manuscript.

In nine patients (7%), gallbladder cancer was incidentally found. We have added this information to Table 1 (page 7).

To better explain the surgical approach in patients with gallbladder carcinoma at our hospital, we have added the following sentences to the Materials and Methods section (page 2): “In case of GBC suspicion, an open radical cholecystectomy was performed, combined with frozen section biopsy of the gallbladder. If frozen section biopsy was positive, without signs of disseminated disease, a lymph node dissection of the hepatoduodenal ligament and a wedge resection of the gallbladder bed were performed. A similar approach was used for patients referred with an incidentally found GBC, after previous cholecystectomy in the referring hospital and after excluding disseminated disease on postoperative CT.”

We agree with the reviewer that preoperative CT should be considered an option to select the most optimal treatment strategy and should guide the type of hospital in which the patient should be treated, as well as the appropriate surgical team. This has also been acknowlegded in the final part of the Discussion on page 10 by the following sentence: “Thereby, not only could patient care and long-term survival outcomes be improved, but also more efficient use of scarce highly specialized hepatobiliary health care resources might be obtained.”

Reviewer 2 Report

The hypotheses that a CNN can distinguish between GBC and benign gallbladder diseases, and that a CNN can use valuable information from adjacent liver parenchyma to improve GBC diagnosis, were tested in this study. The CNN trained by CT scans including both the gallbladder and a rim of adjacent liver parenchyma yielded the best performance in differentiating between GBC and benign gallbladder disease. This study is interesting, well-written and the methodology seems appropriate. 

Author Response

We would like to thank the reviewer for his/her kind comments and for the time invested in reviewing our manuscript.

Reviewer 3 Report

Overall a very interesting manuscript good enough for publication. Just a few minor things.

1. You should explain in either the introduction or materials and method section what CNN is and how exactly it works to help people not familiar with it

2. You should also propose how theoritically the sensitivity and specificity of CNN could improve

Other than that I think you have a very interesting study

Author Response

We would like to thank the reviewer for the kind comments and the time invested in reviewing our manuscript.

In answer to the comments:

  1. We have tried to better explain what a CNN is and how it works, please see our revisions of the Materials and Methods section, page 5.
  2. To theoretically increase the sensitivity and specificity of the CNN, a larger and more heterogeneous dataset should be used. This has been added to the Discussion on page 10.